# Medical Liability of the Vaccinating Doctor: Comparing Policies in European Union Countries during the COVID-19 Pandemic

**DOI:** 10.3390/ijerph19127191

**Published:** 2022-06-11

**Authors:** Carlotta Amantea, Maria Francesca Rossi, Paolo Emilio Santoro, Flavia Beccia, Maria Rosaria Gualano, Ivan Borrelli, Joana Pinto da Costa, Alessandra Daniele, Antonio Tumminello, Stefania Boccia, Walter Ricciardi, Umberto Moscato

**Affiliations:** 1Section of Occupational Health, Department of Health Science and Public Health, Università Cattolica del Sacro Cuore, Largo Francesco Vito 1, 00168 Rome, Italy; carlotta.amantea01@icatt.it (C.A.); mariafrancesca.rossi01@icatt.it (M.F.R.); ivan.borrelli@unicatt.it (I.B.); alessandra.daniele02@icatt.it (A.D.); antonio.tumminello01@icatt.it (A.T.); umberto.moscato@unicatt.it (U.M.); 2Department of Women, Children and Public Health Sciences, Fondazione Policlinico Universitario Agostino Gemelli IRCCS, Largo Francesco Vito 1, 00168 Rome, Italy; 3Section of Hygiene, Department of Health Science and Public Health, Università Cattolica del Sacro Cuore, Largo Francesco Vito 1, 00168 Rome, Italy; flavia.beccia01@icatt.it (F.B.); stefania.boccia@unicatt.it (S.B.); walter.ricciardi@unicatt.it (W.R.); 4Department of Public Health Sciences and Paediatrics, Università di Torino, Via Giuseppe Verdi 8, 10124 Torino, Italy; mariarosaria.gualano@unito.it; 5EPI Unit, Instituto de Saúde Pública, Universidade do Porto, Rua das Taipas, n° 135, 4050-600 Porto, Portugal; joanapc23@gmail.com

**Keywords:** medical liability, COVID-19, vaccination, health policy, European countries

## Abstract

In 2020, the COVID-19 pandemic exhausted healthcare systems around the world, including European Union countries, with healthcare workers at the frontline. Therefore, new health laws and policies have been introduced at the national level in order to offer greater legal protection for health workers. Since the introduction of COVID-19 vaccination, it has led to the development of specific laws to define the compulsoriness for particular categories. This review aimed to evaluate the system of medical liability, focusing on the ten countries of the European Union with the highest rate of vaccination coverage against SARS-CoV-2. A country-by-country analysis was conducted on the different medical liability systems of individual professionals, in general, and with specific focus on the vaccinating doctors. Additional search was conducted to investigate which European states have introduced specific policies in this field, to identify the implementation of any new laws alongside the COVID-19 vaccination campaigns, and to assess which countries have adopted the European Digital COVID Certificate and funded specific compensation programs for COVID-19 vaccination. Our results highlight an extremely fragmented European scenario; therefore, this work could be a starting point to define a common approach for medical liability and related policies in the COVID-19 pandemic.

## 1. Introduction

The COVID-19 pandemic has strained healthcare and managed-care systems around the world, including European Union (EU) countries; despite the expansion of the hospital network, intensive care units, and the activation of temporary hospital areas, the healthcare systems were not ready to handle such a high influx of patients. As of 16 November 2021, Europe is, after the Americas, the second continent most affected by SARS-CoV-2, with more than 5,104,899 deaths and more than 253,640,693 confirmed cases [1].

As reported by the European Commission “the measures needed to contain the pandemic and save lives have a huge impact on people’s livelihoods, their jobs, and their freedoms. Healthcare workers were at the forefront, working day and night to care for COVID-19 patients. EU Member states turned from unilateral measures to support each other, either by receiving COVID-19 patients from neighboring countries or sending healthcare professionals and key medical equipment [2]”.

Beyond the health and human tragedy of SARS-CoV-2, it is now widely recognized that the pandemic triggered the most serious economic crisis in a century [3].

In such a context, the introduction of COVID-19 vaccines acted as a game-changer, despite the vaccine hesitancy issue [4,5]. Vaccines are cost-effective medical tools for disease prevention and the increase in life expectancy [6], and since their introduction, vaccination against COVID-19 has proved to be the best strategy for containing the spread of the virus and for enabling humanity as a whole to overcome the pandemic emergency [7], helped by the adoption of hygienic measures and limitations imposed on citizens [8]. On 27, 28, and 29 December 2020, the vaccination campaign against COVID-19 started across all EU countries simultaneously, and these days were, therefore, dubbed “vaccination days [9]”.

Consequently, to promote vaccination and favour the economic recovery, the European Digital COVID Certificate was introduced, commonly known as the Green Pass, proposed by the European Commission in order to facilitate the safe movement of EU citizens during the COVID-19 pandemic. It is important to underline that vaccination is not mandatory to obtain the Green Pass, as it can also be obtained through a negative SARS-CoV-2 swab conducted within the last 48 (rapid swab) to 72 (molecular swab) hours, or by a certificate stating recovery from COVID-19 [10].

Medical liability is one of the most discussed issues within the current setting of healthcare strain, including the field of vaccinations. Health professionals have had to revolutionize their work by showing exceptional competence in dealing with an extraordinarily critical reality and important scientific uncertainty. Widely celebrated in the first months of the alert in 2020, they have moved on from being defined as ‘heroes’ to ‘defendants’ of this tragic world crisis. The legal and healthcare systems during the pandemic were confronted with something completely new, for which there are still no proven therapies, and, as d’Aloja et al. pointed out, medical and legal science do not always have common goals. The authors believe that “the actionable goal of medicine, although unattainable in concrete terms, is an in-depth knowledge of causes of a disease, its treatment and/or its prevention.” An objective that is difficult to accomplish because one must take into account the error scientiae, i.e., the error resulting from the uncertainty of medical science (notions that are currently held to be valid may no longer be held to be so in the future). This is what occurred during the COVID-19 pandemic; the scientific efforts deployed by the international community were sometimes ineffective, and no clear procedures are currently available to eradicate the risk of infection [11]. This is the reason why some institutional organisations around the world have provided different levels of medical liability protection, even if the latter was rather limited [12].

The legal system of medical liability is articulated on civil, criminal, and disciplinary law codes, based on a model shared by European countries. COVID-19 vaccination has led to the introduction and implementation of specific laws to define the compulsoriness for specific subpopulations. This has repercussions on health professionals; therefore, compensation systems for vaccine damage have been established. The professional liability of healthcare workers, specifically of the vaccinating doctors, is inextricably linked to the issue of informed consent, which, if violated, can result in damage to the patient’s right to information. Informed consent is the fundamental prerequisite for legitimising medical and surgical treatment, and allows the patient to decide freely, autonomously, and after receiving adequate information whether to consent or to refuse medical treatment [13]. The violation of this obligation results in both criminal and civil medical liability [14]. Concerning COVID-19 vaccination, the collection of informed consent is the main stage of medical liability, as the vaccinating physician assesses the patient’s suitability for vaccination, explains the procedure, the adverse events, and exposes the benefits and risks related to the vaccination practice. In the current panorama of the continuous implementation of scientific knowledge and related regulations concerning COVID-19 vaccination, for consent to be truly informed, it is necessary for the vaccinating physician to be punctually updated on the content of the latest scientific knowledge made available by the competent authorities. For example, in Italy, a special law (L. 76 28 May 2021) has been issued to protect the healthcare professional on this issue [15].

This narrative review aimed to evaluate the system of medical liability, both in general and with specific regard to the vaccinating doctor, focusing on the ten countries of the European Union with the highest rate of vaccination coverage against SARS-CoV-2. The research was conducted in order to investigate if and which European states have introduced specific policies in this field and to identify the implementation of any new laws with regard to the COVID-19 vaccination campaigns.

## 2. Methods

For reasons of feasibility, we restricted this study to the ten EU countries with the highest rates of COVID-19 vaccine coverage, according to the last available data on https://covid19.who.int/ (accessed on 16 November 2021). The countries analysed in this study were hereby listed by coverage order: Portugal, Malta, Spain, Italy, Denmark, Ireland, Finland, France, Belgium, and The Netherlands.

To serve the objective of this narrative review, we adopted a three-step research methodology, evaluating scientific and grey literature, and web-screening of institutional national repositories.

Firstly, we performed a literature search of PubMed database, from March 2020 to October 2021, retrieving published articles addressing medical liability with a focus on laws or policies issued specifically to address this theme during the COVID-19 pandemic in EU countries. 

We used the following search query: (“medical liability” OR “medical negligence” OR “medical malpractice”) AND (law* OR legislation OR policy OR policies). We restricted our search to only include the English language but did not apply any other restrictions. Additionally, to retrieve further pertinent publications, we performed a backward reference search considering the eligible articles.

Secondly, for the grey literature research, we used Google search engine (using the same research terms) and the mapping was further extended at the European and EU Member state level, including information on COVID-19 impact on national policies.

Thirdly, to retrieve additional documents missed by the previous methods, we performed desk research on institutional online repositories exploring the European Union, European Commission, and national health ministries or institutes in the selected countries, looking also for supplementary information about vaccine compensation programs and the adoption of the European Green Pass or equivalent certification. We reported the key findings through a set of country-specific profiles that outlined any presence of legislative frameworks towards medical liability and how these frameworks were modified during the COVID-19 pandemic.

The narrative review methodology allowed the researchers to screen grey literature and institutional repositories for information that would not be available on scientific databases, in order to include all relevant information available that fit in within the scope of this research.

## 3. Results

The results were reported by country profiles in terms of the regulatory frameworks for medical liability, vaccination coverage rates, mandatory vaccination, and compensation schemes. Data about vaccination coverage rates for countries, Green Pass adoption, compulsory vaccination, and compensatory programs are reported in Table 1.

In **Portugal**, medical liability is regulated by general legislation about civil liability [16]. According to the Portugal Civil Code, the claimant in damages actions has to prove fault or negligence and, therefore, demonstrate the defendant’s unlawful conduct. Article 483 paragraph two of the Portuguese Civil Code states that “objective liability”, meaning liability without fault or negligence on de defendant’s part, “may only occur when expressly stated in the law (e.g., producer liability)”. Article 487 of the Portuguese Civil Code states that liability arises when “conscious negligence” subsists, meaning the defendant has knowingly committed an unlawful act, even without the intention to cause harm, and is, therefore, obliged to reparation costs to the full extent of the damages that were caused. Finally, article 494 of the Portuguese Civil Code states that if “conscious negligence” does not apply, mere negligence reparations may be fixed at a lower level than the actual damages that were caused [17]. Medical responsibility in Portugal, despite being mainly civil, is also regulate by criminal law when the body, health, or life of a person is placed in danger by a medical professional in the course of a medical act. Criminal liability pertains to specific offenses (i.e., false medical certificate, refusal of medical treatment, etc.); medical acts per se are not considered criminal offences when performed by a qualified person, since they are justified by the patient’s consent and by the therapeutic aim, and the healthcare professional is criminally liable only when acting against the patient’s consent or when performing an unnecessary act that endangers the patient’s life. Finally, disciplinary responsibility (professional or administrative) is resolved extrajudicially [17].

Despite holding the record of doses administered and with 87% of the population being fully vaccinated, Portugal has not introduced any policies concerning medical liability regarding anti-SARS-CoV-2 vaccines. Vaccination in Portugal is on a voluntary basis only and is not mandatory for healthcare professionals [18].

The law of **Malta** assesses medical liability under both the penal and civil code. According to the penal code, healthcare professionals can be charged with a criminal offence, the same way as other citizens. Regarding the damages derived from the practice of a medical profession, Article 1033 (Chapter 16—Laws of Malta) states that: “Any person who, with or without intent to injure, voluntarily or through negligence, imprudence, or want of attention, is guilty of any act or omission constituting a breach of the duty imposed by law, shall be liable for any damage resulting therefrom [19].” Despite being the second European country in terms of vaccination doses administered and with 86% of the population being fully vaccinated, the Maltese government has not introduced specific policies for health professionals, nor has it modified medical liability regimes with regard to vaccination against COVID-19. The administration of vaccines is on a voluntary basis only; therefore, there is no specific compensation policy for SARS-CoV-2 vaccines. Maltese healthcare professionals are not subjected to mandatory vaccination.

Medical liability in **Spain** is based on the fault system. However, in the case of the direct liability of the National Health System, the possibility of establishing compensation derives from the normal administrative activity and, consequently, from the activities of professionals working there. The no-fault model does not exist. The responsibility lies with the administration, which, however, can seek recourse against the healthcare professional responsible for the damages [20]. More specifically, two different legal regimes apply depending on whether the medical malpractice episode occurs in the public or private healthcare sector [21,22]. If the injury occurs in the course of the provision of medical services by a private professional, a private law regime, namely, the civil code [23], and the general consumer and user protection law [24], governed by civil law, would apply. However, if the accident occurs under the public regime, liability would be governed by the public-sector legal regime law [25], falling under administrative law [26]. In the event that the conduct in question constitutes a crime, liability is acquired not by the medical facility, but by the physician himself (and the public authority itself would be liable on a subsidiary basis) according to the dictates of the criminal code [27]. Spain does not have compulsory vaccinations. Regarding COVID-19, the Spanish COVID-19 vaccination strategy explicitly adopted a model of voluntary vaccination. However, recently, the Parliament of the Autonomous Community of Galicia passed Law 8/2021 on 2 April 2021, amending the regional Health Act of Galicia; it aims to introduce administrative fines for the unjustified refusal of vaccination. Nonetheless, in Spain, there is no obligation to get vaccinated for any diseases. To accomplish Article 43 of the Spanish Constitution about the right to health protection, the mean of Article 3 of Organic Law 3/1986 was allowed in part by the Spanish government to impose vaccinations only in a few selected cases. Law 8/2021 considers the refusal to comply with a vaccination order as a minor administrative offence, which can be punished with an administrative fine ranging from EUR 1000 to 3000 (Article 41bis(d)). However, it can also amount to “a serious or very serious administrative offence, with a corresponding fine of up to EUR 60,000 or EUR 600,000 respectively, if the vaccine refusal ‘could pose a risk of serious or very serious harm to the health of the population” (Articles 42bis(c) and 43bis(d)). It is necessary to point out that a framework for compulsory vaccination could only be issued by organic laws. Therefore, laws such as Law 8/2021 could be interpreted as unconstitutional in substantive terms, according to the principles of legality, adequacy, necessity, and proportionality. The unconstitutionality appeal has recently been declared admissible by the Spanish Constitutional Court, which has also ordered the interim suspension of the contested provision. In Spain, the share of people fully vaccinated against COVID-19 amounts to 80%, whereas in healthcare workers, 98% was reached [28,29].

The criminal and civil medical liability system in **Italy** was revolutionised by the Gelli–Bianco Law (L. 24/17), in Articles 6 and 7, respectively. Article 6 introduced into the Criminal Code Article 590-sexies establishes non-punishability for a healthcare worker who commits an act constituting manslaughter or culpable personal injury, and if the event occurred due to a minor fault caused by inexperience. This applies if the guideline recommendations as defined and published in accordance with the law were followed or, in the absence of these, if good clinical and care practices, provided that they are appropriate to the case, have been complied with. Article 7 defines a two-track system of civil liability, which is more favourable to the healthcare professional. The doctor is liable for noncontractual liability (Art. 2043 of the civil code), whilst the facility is liable for contractual liability (Art. 1218, 1228 of the civil code). The main differences between contractual and noncontractual liability relate to three specifics: the burden of proof; the statute of limitations for compensation; and compensable damage. In the case of contractual liability, the burden of proof is on the hospital; the statute of limitations is ten years, and compensation is limited to that foreseeable damage at the time the obligation was incurred. In the case of noncontractual liability, the burden of proof is on the patient; the statute of limitations is five years, and the indemnifiable damage is potentially unlimited. The different liability regime tends to offer greater protection to the healthcare workers, since it induces the patient to take action against the hospital rather than the physician. During the COVID-19 pandemic, due to the exceptionally critical and uncertain conditions in which health workers worked, Law 76/21 of 28 May was adopted to protect health workers. Article 3 of Law 76/21 states that legal punishability is not applicable for vaccinating doctors who have caused manslaughter (Article 589 of the criminal code) or culpable personal injury (Article 590 of the criminal code) to a patient, if they complied with the instructions contained in the marketing authorisation issued by the competent authorities and the circulars published on the institutional website of the Italian Ministry of Health relating to vaccination activities; Article 3*bis* also states the exclusion of punishability for all healthcare professionals who have committed manslaughter or culpable personal injury in the exercise of the health profession in emergency conditions, punishable only in cases of gross misconduct.

According to Italian jurisprudence, the attribution of the degree of guilt is a function of the specific subjective profile concerning the agent: the more adequate the subject is to observe the rule and the greater the reliance of third parties, the greater the degree of guilt. Therefore, the quantum of demand for the observance of the precautionary rules is an important factor in the graduation of guilt [30]. Article 4 makes vaccination compulsory for all healthcare workers: in the event of noncompliance, they risk suspension from their profession if there is no possibility of being assigned to a job position that does not entail any risk of spreading contagion [15]. Regarding informed consent (regulated in Italy by Law 219/17), Law 76/21 also guarantees legal protection for the vaccinating physician. In the collection of consent to vaccination, a point of care takes place, whether consent is given or refused. In fact, should the patient reject the vaccination, the doctor is in any case obliged to investigate the reasons for the refusal. In addition, in Italy, there is Law n. 210 of 25 February 1992, which provides compensation to persons irreversibly injured by vaccinations, transfusions, and the administration of infected blood products [31]. Italy ranks fourth among the most vaccinated countries in the EU, with 73% of the population having completed the vaccination scheme by 16 November 2021 [32].

**Danish** medical liability is divided into three main fields: civil, criminal, and disciplinary. In criminal law, the doctor is liable in the event of severe fault, while in civil law, most medical disputes are resolved. In Denmark, there is no procedure to be followed in the hypothesis of compensation for biological damages, even if the method used is obviously uniform. The extent of compensation is linked to the extent of the injuries sustained and the consequences they have on the life of the injured party. The starting point for compensation for personal injury is always the medical opinion of the specialist, as established by the Damages Act [33]. The Danish legal system has not introduced specific laws or policies to protect the vaccinating physician. COVID-19 vaccination is offered free of charge and on a voluntary basis to the eligible population. Denmark ranks as the fifth country among the countries of the European Union that have administered higher vaccine coverage rates, with 76% of the population having been fully vaccinated [34].

The **Irish** legal system is based on common law, a mainly jurisprudential model. As far as civil responsibility is concerned, medical culpability subsists if the doctor violates the duty of care. The duty of care is not based only on a contractual duty, but it applies any time a request of care is made [35]. Medical negligence is assessed using the so-called “Bolam test”, regarding the nature of the professional activity that was exercised and the average knowledge that could be required from the doctor [36]. Furthermore, doctors have a duty to refresher courses. When the legal action is between the doctor as a public employee and the patient, it is addressed by the National Health Service. The doctor is not directly involved in the process and is only liable disciplinarily. On the contrary, when the doctor works in the private sector, the general legal principles apply. As far as penal responsibility is concerned, healthcare workers can be liable for major culpability. Ireland has vaccinated 78% of its population; no mandatory SARS-CoV-2 vaccination policy for healthcare professionals has been established. The vaccination is administered on a voluntary basis only [37].

Medical liability in **Finland** is represented by a system based on no-fault compensation, which means that compensation is paid regardless of who is at fault. The doctor remains liable on a disciplinary basis. The patient only has to prove that the harmful event was foreseeable and avoidable, and the burden of compensation falls on the public structure, which only awards damages that are strictly consequential to the event. However, healthcare workers are obliged to take out public insurance and, thus, contribute to the compensation fund [35]. The COVID-19 vaccine is available to the population on a voluntary basis, including healthcare workers [38]. In total, 72% of Finland’s population has been fully vaccinated. Finally, no specific laws or policies have been introduced regarding medical liability for COVID-19 vaccination.

In **France**, healthcare professionals are liable for their actions in administrative, civil, and criminal law, depending on the offence committed. Regarding COVID-19 vaccination, damages caused by the vaccine are compensated through a national compensation system; vaccinating physicians are rarely personally liable. Civil liability concerns doctors working in a private structure, while administrative liability concerns doctors working in a public structure. The Kouchner Law of 4 March 2002 maintains the fault-based system, but introduces a public compensation fund for incidents that are not directly attributable to the negligence of someone, such as compulsory vaccination or nosocomial infections. However, compensation is only granted in the event of serious consequences by the “Office National d’Indemnisation des Accidents Médicaux, des affections iathrogènes [39] nosocomiales (ONIAM)”, which provides civil compensation, based on national solidarity [40,41,42]. Vaccination is compulsory for French healthcare workers, according to article 12 of Law n. 2021-1040 of 5 August [43,44,45,46]. On 16 November 2021, 69% of the population completed the vaccination scheme. The French government did not change the medical liability regimes regarding vaccination against COVID-19, namely, because compensation for damages would be covered by national solidarity through ONIAM (Law 4, March 2002).

In **Belgium**, medical liability is based on the doctor’s fault. The Regional Commissions of Conciliation and Indemnification (CRCI) decides, on a regional basis, on medical injury disputes, iatrogenic or nosocomial infection disputes, and any other disputes between patients and healthcare professionals, healthcare-providing structures, or healthcare supplies production companies. As far as penal responsibility is concerned, minor culpability is also punishable. No specific laws were introduced in Belgium concerning healthcare professionals during the COVID-19 pandemic, and the medical liability laws were not modified. Incentive bonuses were granted by the INAMI (Institut national d’assurance maladie-invalidité, i.e., the National Institute for Illness and Disability Insurance) for healthcare professionals who exercised their profession during the pandemic [47]. Seventy-five percent of Belgium’s population is fully vaccinated. No compulsory vaccination laws were introduced.

In the legal system of **The Netherlands**, to prove malpractice, a plaintiff must establish that the physician undertook a duty to treat or care for a patient, that the physician failed to comply to the standard of care (as evidenced by expert witnesses or obvious errors), that the breach of duty was a proximate cause of injury, and that damages were incurred [48]. In the field of medical liability, no new laws have been introduced to protect the health professional regarding COVID-19 vaccination, which is available to everyone and is not compulsory. The Netherlands is the tenth and final country in the European Union for its vaccination coverage rate, with 73% of the population being fully vaccinated.

**Table 1 ijerph-19-07191-t001:** Ten European countries with the highest vaccination coverage rates.

Country *	Doses Administered at 16/11 (Per 100 People)	Compulsory Vaccination	Note	Green Pass	% Population Fully Vaccinated	Vaccination Compensation	Laws/Policies (Compulsory Vaccination)
**Portugal**	158	No		Yes	87%	No	
**Malta**	181	No		Yes	86%	No	
**Spain**	156	No		No	80%	No	
**Italy**	153	Yes	Only for Healthcare Professionals	Yes	73%	No	Artt. 3,4; L.76 of 28 May 2021 [49]
**Denmark**	153	No		Yes	76%	No	
**Ireland**	149	No		Yes	76%	No	
**Finland**	149	No		No	72%	No	
**France**	151	Yes	Only for Healthcare Professionals	Yes	69%	Yes (ONIAM)	Art. 12; L.2021-1040 of 5 August [46]
**Belgium**	148	No		Yes	75%	No	
**The Netherlands**	140	No		No	73%	No	

* Countries are sorted in descending order by vaccination coverage rate (data from: COVID World Vaccination Tracker—The New York Times. Available online: https://www.nytimes.com/interactive/2021/world/covid-vaccinations-tracker.html (accessed on 16 November 2021)).

## 4. Discussion

Overall, medical liability is similar across Europe, and responsibility is both civil and penal in most of the countries included in the review. Small differences were identifiable in the specific laws according to which medical liability cases were evaluated.

In six out of the ten countries taken into account (Malta, Spain, Italy, France, Belgium, and the Netherlands), medical liability is based on the fault system. Ireland has a similar system; however, it is represented by the common law, a mainly jurisprudential model, and not the civil law system as in the other nine countries analysed. In Portugal, medical liability is regulated mainly by general legislations about civil liability, and in Finland is represented by a system based on no-fault compensation, which means that compensation is paid regardless of who is at fault. The fault-based system creates a dual liability profile for the doctor, both in criminal and civil law, compared with the no-fault system, in which the administration is liable for the damages. In this context, Finland is a European Union country with the most liability protection for the physician, out of the ten analysed, as far as the COVID-19 scope is concerned.

Some of the highlighted differences between the medical liability systems across Europe could be tied to the differences in the healthcare systems the selected countries had. Furthermore, it is important to consider that the COVID-19 pandemic might have posed a significant additional burden on healthcare systems, exacerbating their frailties. Therefore, most of the laws across the EU, emanated since the pandemic started, have aimed to improve the efficiency of healthcare systems and to safeguard healthcare workers by enforcing recruitment plans, mandatory vaccinations, and liability shields.

Among the screened countries, the only one with a compensation fund specific for the SARS-CoV-2 vaccine was France, while Italy was the only state where a law has been introduced to redefine the limits and applicability of medical liability during the pandemic.

In Italy, the approval of Law n.76 of 28 May 2021 was decided on with the intention of finding a balance between the protection of the vaccinating user and greater guarantees for the vaccinating physician [15].

In the early phases of the pandemic, medical care shifted from a patient-centred approach to being guided by public-health duties [50]. Italy was the first country hit in Europe, and faced the consequences of dramatic shortages of ICU beds, respirators, and trained ICU staff [50,51]. The necessity of having to work in an emergency condition and the pressure exerted by the pandemic on healthcare systems may have favoured the spread of defensive medicine [52,53]. This approach, applied in the context of vaccination, would have weakened the vaccination campaign, as it has emerged in some cases of adverse events to vaccination; therefore, the legislature in Italy aimed to support the vaccination campaign, the medical class (already tried by a year of the pandemic), and the health of the population by inserting legislative protections. Indeed, legal protection for physicians would reduce the stress they were under, improving the quality of care. It would be good to remember though that the ultimate purpose of liability protection is patient protection [54].

Within a similar scope, France introduced a specific compensation fund for COVID-19 vaccination as part of measures implemented during the first two waves of the pandemic to find a balance between the pandemic’s impact and the response to it [55].

One possible reason why the example set by Italy and France was not adopted by other countries may rely on the extreme burden faced by these two countries in comparison to the other considered European countries.

As for mandatory vaccinations, this specific policy was the main strategy to prevent hospitalizations and deaths due to COVID-19, and helped to reduce infections in healthcare workers, which, in turn, would reduce the risk in hospital patients. Despite this virtuous cycle, only two countries included in this review, Italy and France, adopted mandatory vaccinations for healthcare workers early on. Most of the countries included in the review did not adopt mandatory vaccination; this might be due to the perception that mandatory vaccination is against national law policies and The Constitution. Most countries in the EU have mandatory vaccination for children, but have not adopted mandatory COVID-19 vaccination. Despite this, the introduction of the European Green Pass, or equivalent certifications, has induced a large portion of the general population, in particular vaccine-hesitant people, to get vaccinated in order to have access to their workplace, leisure activities (restaurants, bars, cinemas, theatres, etc.), public transport, training centres, etc.

Pertaining to the Green Pass certification, 8 out of the 10 screened countries adopted the European Green Pass, or an equivalent certification; Spain and Finland did not adopt any form of mandatory certification.

Despite the differences in the specific laws, some uniformity may be observed pertaining to medical liability in EU countries, by shifting the focus on civil liability as the main healthcare professionals’ liability. Another strategy may be to make European certifications mandatory; for example, the EU Green Pass was not mandatory in all EU countries, albeit making it so many contributed to uniform vaccination policies and, in turn, vaccination healthcare professionals’ medical liability. Homogeneous medical liability policies may eventually be achieved, but a joint effort would be required both at a European level and at a single-country level. Furthermore, the creation of a supranational no-fault compensation programme for COVID-19, such as the COVAX programme for low- and middle-income countries, at the European level, perhaps with the involvement of the WHO European office, could also help to provide additional protection for patients and to standardise the response across Europe.

## 5. Conclusions

Our results highlighted an extremely fragmented European scenario; therefore, this work could be a starting point and a useful tool for policy makers at the EU level to define common approaches and standards for medical liability and related policies in the COVID-19 pandemic, or in future medical emergency settings. Finally, this work could summarize all strengths and weaknesses of different EU countries, highlighting how to integrate the legal liability systems in single countries.

## Data Availability

Not applicable.

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
