# Peer review of "Medical Liability of the Vaccinating Doctor: Comparing Policies in European Union Countries during the COVID-19 Pandemic"

_ijerph, 2022, doi:10.3390/ijerph19127191_

Round 1

Reviewer 1 Report

I read with great interest the work entitled  ‘Medical liability of the vaccinating doctor: comparing policies in European Union Countries during the COVID-19 pandemic”.

I ask the authors some  fundamental clarifications to make the paper publishable.

Major Concerns

a)      The first note concerns the introduction. It should be further emphasized that the legal and health systems during the pandemic found themselves facing something entirely new for which there are no therapies to date for which there is evidence of effective efficacy. [cfr d'Aloja, E., et al. "COVID-19 and medical liability: Italy denies the shield to its heroes." EClinicalMedicine 25 (2020).].

b)     The methods are described very roughly. It looks like a mix between scientific research and internet research. The number of articles considered is not clear, the selection criteria are not clear nor why it was necessary to resort to other sources. The above does not make the results described by the authors reproducible. It is suggested that the paper be rewritten as a 'narrative review'.

c)      The types of damage included in the research do not include that of damage to the right to information. This data is assessed in the course of the medical liability assessment. Are there specific laws about it? How was the problem of the representation to the patient of the existence of 'error scientiae' in the various countries faced?

d)     As regards Portugal, the data concerning medical liability in the criminal code should be reported.

e)     As far as Spain is concerned, it should be clarified whether we are talking about the civil or criminal code.

f)       As for Italy

1) it should be clarified whether the code provides for anything regarding informed consent.

               2) The definition of gross negligence should be given.

               3)  Reference should be made to the provisions of Italian jurisprudence for compulsory vaccines for the general population. (i.e. L   210/1992). This topic deserves to be deepened also during the discussion.

               g)  The article should explain to non-European readers (perhaps with the help of a table) the difference in legal terms between a mandatory vaccination required by law and an 'optional' but necessary one to obtain the Green Pass.

h) The conclusions reached by the authors and the hopes are utopian compared to the concrete situation they described in the discussion. In the article the authors should explain how in their opinion it would be possible in practice to achieve the desired uniformity.

i) The article, theoretically focused on medical liability for vaccinating doctors, actually describes states in which it was not deemed necessary to protect this category. In the discussion one should at least hypothesize what are the reasons why Italy and France have deemed it necessary to introduce greater protections while the other states do not.

i) The number of references should be increased.

Author Response

Please see the attachment, for the manuscript with revisions.

Response A

Thank you for your valuable suggestion. The following paragraphs has been added to the introduction: “The legal and healthcare systems during the pandemic were confronted with something completely new, for which there are still no proven therapies, and, as the authors d'Aloja et al. point out, medical and legal science do not always have common goals. The authors believe that “the actionable goal of medicine, although unattainable in concrete terms, is an in-depth knowledge of causes of a disease, its treatment and/or its prevention.” An objective that is difficult to accomplish because one must take into account the error scientiae, i.e. the error resulting from the uncertainty of medical science (notions that are currently held to be valid may no longer be held to be so in the future). This is what occurred during the COVID-19 pandemic, the scientific efforts deployed by the international community were sometimes ineffective, and no clear procedures are currently available to eradicate the risk of infection.”

“This is the reason that some institutional organisations around the world have provided different levels of medical liability protection, even if the latter was rather limited.”

(D’Aloja E, Finco G, Demontis R, et al. COVID-19 and medical liability: Italy denies the shield to its heroes. EClinicalMedicine; 25. Epub ahead of print August 1, 2020. DOI: 10.1016/J.ECLINM.2020.100470).

(Napoli PE, Nioi M, D’aloja E, et al. Safety recommendations and medical liability in ocular sur-gery during the covid-19 pandemic: An unsolved dilemma. Journal of Clinical Medicine; 9. Epub ahead of print May 1, 2020. DOI: 10.3390/JCM9051403)

Response B:

In thanking you for your input, we have clarified that this study is a narrative review properly within the text. This review is not framed as a systematic review because we agree with you that only searching databases for scientifics articles would lead to disregard relevant information (relevant articles that are country-specific may not be in English, publishing takes time and therefore scientific literature may not be up to date, etc.); a systematic review would not be able to produce relevant results because topics such as EU Green Pass, vaccination coverage, mandatory vaccination policies, and so on, may not be up to date in scientific repositories. For this reason, a narrative review was performed, as explained in the method section: “[...]  performing scientific and grey literature, and web-screening of institutional national repositories. " We have added a sentence to further explain the reason why other sources were necessary: " The narrative review methodology allowed the researchers to screen grey literature and institutional repositories for information that would not be available on scientific databases, in order to include all relevant information available fitting within the scope of this research."

Response C:

Yes, damage to the right to information was not included among the variables in this narrative review. Thank you for the interesting suggestion included in the introduction: “The professional liability of the healthcare workers, and specifically of the vaccinating doctor, is inextricably linked to the issue of informed consent, which, if violated, can result in damage to the patient's right to information. Informed consent is the fundamental prerequisite for legitimising medical and surgical treatment and allows the patient to decide freely, autonomously, and after adequate information, whether to consent or to refuse medical treatment. Violation of this obligation results both criminal and civil medical liability. Concerning COVID-19 vaccination, the collection of informed consent is the main moment of medical liability, as the vaccinating physician assesses the patient's suitability for vaccination, explains the procedure, the adverse events, and exposes the benefits and risks related to the vaccination practice. In the current panorama of continuous implementation of scientific knowledge and related regulations concerning COVID-19 vaccination, for consent to be truly informed it is necessary for the vaccinating physician to be punctually updated on the content of the latest scientific knowledge made available by the competent authorities. For example, in Italy a special law (L. 76 28 May 2021) has been issued to protect the healthcare professional on this issue.”

Response D:

Thank you for pointing out that we could have elaborated more on criminal liability in Portugal, a paragraph has been added concerning criminal and disciplinary liability: “Medical responsibility in Portugal, despite being mainly civil, is also regulate by criminal law, when the body, health or life of a person by a medical professional, in the course of a medical act. Criminal liability pertains to specific offenses (i.e.: false medical certificate, refusal of medical treatmet, etc); medical acts per-se are not considered criminal offences when performed by a person qualified for it, since they are justified by the patient’s consent and by the therapeutic aim, and the healthcare professional is criminally liable only when acting against the patient’s consent or when performing an unnecessary act that endangers the patient’s life. Finally, disciplinary responsibility (professional or administrative) is resolved extra-juridically."

Response E:

Thank you for pointing out the lack of clarity related to the Spanish medical liability system. We added the following parapgraph to clarify where civil, criminal, or administrative law jurisdiction apply. “More specifically, two different legal regimes apply depending on whether the medical malpractice episode occurs in the public or private health care sector. If the injury occurs in the course of the provision of medical services by a private professional, a private law regime, namely the Civil Code, and the General Consumer and User Protection Law, governed by civil law, will apply. However, if the accident occurs under the public regime, liability will be governed by the Public Sector Legal Regime Law, falling under administrative law. In the event that the conduct in question constitutes a crime, liability is acquired not by the medical facility, but by the physician himself (and the public authority itself will be liable on a subsidiary basis) according to the dictates of the Criminal Code.”

However, considered the peculiarity of the spanish healthcare system (all Spanish inhabitants enjoy the provision of medical assistance, which is mostly financed by taxation), medical liability mostly relies on administrative jurisdiction, as was expressed in the text.

Response F:

Thanks for the clarification, we have integrated: “Regarding informed consent (regulated in Italy by Law 219/17), Law 76/21 also guarantees legal protection for the vaccinating physician. In the collection of consent to vaccination, a moment of care takes place, whether consent is given or refused. In fact, should the patient reject the vaccination, the doctor is in any case obliged to investigate the reasons for the refusal. In addition, in Italy there is Law n. 210 of 25 February 1992, which provides compensation to persons irreversibly injured by vaccinations, transfusions and administration of infected blood products.”

And: “According to Italian jurisprudence, the attribution of the degree of guilt is a function of the specific subjective profile concerning the agent: the more adequate the subject is to observe the rule and the greater the reliance of third parties, the greater the degree of guilt. Therefore, the quantum of demand for observance of the precautionary rules is an important factor in the graduation of guilt.”

Response G:

Thank you for the input. We have added a sentence to further explain that Green Pass did not mean that vaccination was mandatory: " It is important to underline that vaccination was not mandatory to obtain the Green Pass, as it could also be obtained through a negative Sars-CoV-2 swab done within the previous 48 (rapid swab) to 72 (molecular swab) hours, or by a certificate stating recovery from COVID-19". However, we feel like going into detail about the legal differences between mandatory vaccination and mandatory green pass is beyond the scope of this review, as we do not go into details pertaining to the legal aspects of mandatory vaccinations but keep our focus on medical liability concerning vaccination administration.

Response H:

In thanking you for the suggestion, we have expanded on how medical liability policies could be more homogeneous: " Despite the differences in the specific laws, some uniformity may be achieved pertaining medical liability in EU countries, by shifting the focus on civil liability as the main healthcare professionals’ liability. Another strategy may be to make European certifications mandatory; for example, the EU Green Pass was not mandatory in all EU countries, albeit making it so may have contributed to uniform vaccination policies and, in turn, vaccination healthcare professionals’ medical liability. Homogeneous medical liability policies may eventually be achieved but a joint effort would be required both at a European level and at a single country level. Furthermore, the creation of a supranational no-fault compensation programme for COVID, like the COVAX programme for low- and middle-income countries, at the European level, perhaps with the involvement of the European office of the WHO, could also help to provide additional protection for patients and to standardise the response across Europe.”

Response I:

Thank you for this precious input. The review investigated policies put in place to protect vaccinating doctors in 10 EU countries (the selection criteria is reported in the metodology section); it was not known a-priori which countries would have or lack such policies. As per your valuable suggestion, we added a paragraph elaborating on countries where medical liability policies for vaccinating doctors were introduced: " In the early phases of the pandemic, medical care shifted from a patient-centred approach to being guided by public-health duties. Italy was the first country hit in Europe and faced the consequences of dramatic shortages of ICU beds, respirators, and trained ICU staff. The necessity of having to work in an emergency condition and the pressure exerted by the pandemic on health care systems may have favoured the spread of defensive medicine. This approach applied in the context of vaccination, would have weakened the vaccination campaign, as emerged in some cases of adverse events to vaccination, so the legislature, in Italy, aimed to support the vaccination campaign, the medical class, already tried by a year of pandemic, and the health of the population, by inserting legislative protections. Indeed, legal protection for physicians would reduce the stress they are under, improving the quality of care. It would be good to remember that the ultimate purpose of liability protection is patient protection.

With a similar scope, France introduced a specific compensation fund for COVID-19 vaccination, as part of measures implemented during the first two waves of pandemic to find balance between the pandemic impact and the response to it.

One possible reason why the example set by Italy and France was not adopted by other countries may rely in the extreme burden faced by these two countries in comparison to the other considered European countries.

Response I:

Following your valuable advice, we increased the bibliography, expanding the references in the previous text and adding more references in the new sections in response to your previous comments and suggestions. New references added: 4,5,7,8,11,12, 14, 21-27, 50-55.

Reviewer 2 Report

Thank you for the possibility of reading and evaluating this paper.
It is a pretty interesting paper, however, I have a few important suggestions and remarks.
This paper analyses the medical liability systems in the context of COVID-19. However, the pandemic is expiring, so the goal of the paper and conclusions should be improved and strengthened. The novelty should be more accented. The conclusions need better argumentation and policy implications. The Authors only have written that their study is “a starting point” and have not given any suggestions "a starting point" to which/what actions. The Authors have written only general conclusions without any policy implications. They have analyzed the systems and summarized that the EU countries are different, which conclusion is rather obvious.
Please, expand the conclusions and more adequate specify the goal. Maybe, the COVID-19 can be a good background for any recommendations for the future?
Moreover, please correct Table 1. It needs editorial corrections.  I suggest preparing notes (below table) in order to give references to the acts included in the last column.
Please replace big letters with small ones in words like countries, state, pandemic.
‘Now’ in line 44 on page 1 denotes 31.05.2022? Please, correct the style of the sentence.
It is sth wrong with line 104 on page 2 (double spaced?)
The Netherland???? (line 82 page 2)
Please, expand the literature review related to the role of vaccinations

Author Response

Please see the attachment, for the revised manuscript.

Response 1:

In thanking you for the suggestion, we have expanded on how medical liability policies could be more homogeneous: "Despite the differences in the specific laws, some uniformity may be achieved pertaining medical liability in EU countries, by shifting the focus on civil liability as the main healthcare professionals’ liability. Another strategy may be to make European certifications mandatory; for example, the EU Green Pass was not mandatory in all EU countries, albeit making it so may have contributed to uniform vaccination policies and, in turn, vaccination healthcare professionals’ medical liability. Homogeneous medical liability policies may eventually be achieved but a joint effort would be required both at a European level and at a single country level. Furthermore, the creation of a supranational no-fault compensation programme for COVID, like the COVAX programme for low- and middle-income countries, at the European level, perhaps with the involvement of the European office of the WHO, could also help to provide additional protection for patients and to standardise the response across Europe."

Response 2:

Editorial corrections have been made to the layout of Table 1. Furthermore, we have added the proper references to the laws cited in the last column.

Response 3:

Thank you for your suggestions, we have corrected all orthographic and editing errors reported to us.

Response 4:

Thank you, we increased the references number, especially regarding the literature review on the role of COVID-19 vaccination (ref 4,5,7,8).

Round 2

Reviewer 1 Report

Compared to the first version, the paper has significantly improved. I have no other observations to make to the authors.

Reviewer 2 Report

I accept this version of the paper.